# The relational association between multiple sexual partners and HIV testing on cervical cancer screening among women of reproductive age in Ghana: A national population-based study

Agani Afaya[1], Wise Awunyo[1,2]*, Mary Adaeze Udeoha[1], Matilda Mawusi Kodjo[1], Maxwell Tii Kumbeni[3]

1 Department of Nursing, School of Nursing and Midwifery, University of Health and Allied Sciences, Ho, Ghana, 2 Department of Surgery, Surgical Sub-BMC, Ho Teaching Hospital, Volta Region, Ghana, 3 Medicaid Policy and Fee-for-Service Operations, Medicaid Division, Oregon Health Authority, Salem, United States of America

* wiseawunyo38@gmail.com

## Abstract

### Background

Cervical cancer screening remains vital in the early detection of precancerous lesions and promotes better treatment outcomes. Though evidence suggested multiple sexual partners and HIV infection as risk factors for cervical cancer, limited studies have investigated how multiple sexual partners and HIV infection impact cervical cancer screening in Ghana. Therefore, this study assessed the association of multiple sexual partnership and HIV testing on cervical cancer screening among Ghanaian women of reproductive age.

### Methods

The study utilized data from the 2022 Ghana Demographic and Health Survey. A weighted representative sample of 15,014 women from the 16 regions of Ghana was used for the analysis. Descriptive statistics, Pearson's Chi-square and multivariable logistic regressions were used to analyze the data. Adjusted odds ratios (aORs) at 95% Confidence Intervals were presented from the multivariable logistic regression.

### Results

The prevalence of cervical cancer screening was 5.0%. We found lower odds of cervical cancer screening with multiple sexual partners at the bivariate level [aOR=0.69, 95% CI: 0.52–0.90], but no association was found in the multivariable model. On the other hand, women who had ever tested for HIV [aOR=4.73, 95% CI:3.39–6.59] were more likely to screen for cervical cancer than those who had never tested. This was still significant after adjusting for covariates [aOR=2.47, 95% CI:1.61–3.80].

**Data availability statement:** The dataset (Ghana Demographic and Health Survey, 2022) used for the study is publicly available at the Measure DHS repository (https://dhsprogram.com/data/dataset/Ghana_Standard-DHS_2022.cfm?flag=1), and the report is available at https://dhsprogram.com/pubs/pdf/FR387/FR387.pdf.

**Funding:** The author(s) received no specific funding for this work.

**Competing interests:** The authors have declared that no competing interests exist.

**Abbreviations:** aOR, Adjusted Odds Ratio; CI, Confidence Interval; cOR, Crude Odds Ratio; HDI, Human Development Index; HIV, Human Immunodeficiency Virus; HPV, Human Papilloma Virus; ICF, International Coaching Federation; KOICA, Korean International Cooperation Agency; LMICs, Low, and Middle, Income Countries; LMICs, Lower, and Middle, Income Countries; PCA, Principal Component Analysis; PMI, President's Malaria Initiative; PPS, Probability Proportional to Size; SSA, sub, Saharan Africa; UK AID, United Kingdom Agency for International Development; UNFPA, United Nations Population Fund; UNICEF, United Nations International Children's Fund; USAID, United States Agency for International Development; VIF, Variance Inflation Factors; WHO, World Health Organization

## Conclusions

The study highlights the influence of multiple sexual partners and HIV testing on cervical cancer screening uptake among women of reproductive age in Ghana. Though the study found no significant association between multiple sexual partners and cervical screening uptake, HIV testing was a predictor of cervical cancer screening among women in Ghana. We recommend continued creation of awareness of cervical cancer screening among women of reproductive age.

## Background

Cervical cancer continues to be a significant public health concern worldwide, particularly in low- and middle-income countries (LMICs) [1,2]. Globally, cervical cancer is the fourth most common cancer in women, accounting for about 660,000 new cases and around 350,000 deaths in 2022, with 94% of these deaths occurring in LMICs [3–5]. Cervical cancer incidence and death rates show significant regional variations. Countries with a low Human Development Index (HDI) have incidence rates twice as high as mortality rates, five times higher than in high HDI countries [4,6,7]. It is predicted that the worldwide burden of cervical cancer may increase to 760,082 new cases and 411,035 deaths by 2030, reflecting a 14.8% increase in cases and a 17.8% increase in deaths [8]. The World Health Organization (WHO) indicated that cervical cancer is largely preventable, but poor access to prevention and screening leads to 90% of deaths [9].

In sub-Saharan Africa (SSA), cervical cancer is the most frequent disease among women, accounting for 22.2% of all malignancies and ranking as the main cause of cancer-related deaths in the region [10]. In 2022, among the top 20 countries with the highest cervical cancer burden, 18 of them were in SSA. These countries accounted for 23% of total cervical cancer mortality. Additionally, 85% of patients presented at late stages of the disease [3,11] and this proportion has increased over the last two decades, with large annual increases observed in Malawi (7.9%) and Uganda (2.2%) [12]. In 2023, an estimated 3,072 new cases of cervical cancer reported in Ghana, resulting in an age-standardized incidence rate of 27 per 100,000 women. In the same year, an estimated 1,815 women died from the condition, resulting in an age-standardized death rate of 16.9 per 100,000 women. It is anticipated that without any intervention, a total of 329,925 women in Ghana would die from the disease during 2020–2070, rising to 1,144,566 by 2120 [13].

Screening is a vital component in preventing cervical cancer. Beginning at age 30, Human Papillomavirus [14] DNA testing is the main screening technique, according to the WHO's current screening recommendations [1,3]. The standard schedule for cervical cancer is every five to ten years for women of 30 and above till the age of 50 [15]. However, for HIV-positive women, screening should begin at age 25 and be done every three to five years [3,16]. Although cervical cancer screening has been identified to curb the menace, it is challenging, especially in low-resource settings [17]. In 2020, the WHO developed a global strategy to accelerate the elimination

of cervical cancer. The WHO's global strategy 90-70-90 aims to achieve 90% of girls vaccinated against HPV, 70% of women in the world screened using the HPV test, and 90% of women with cervical cancer should receive treatment appropriately by 2030 [18–23]. Countries will be regarded to have eradicated cervical cancer as a public health hazard when rates of new cases fall to 4 per 100,000 women-years. Modelling indicated that if these objectives are attained in 78 LMICs, cervical cancer will be eradicated in all LMICs, and a total of 74.1 million cancer cases and 62.6 million deaths would be prevented over the course of a century [13].

In Ghana, studies have already found low cervical cancer screening among women. A recent study in Ghana found that 7.27% of women aged 30–49 get screened for cervical cancer, with notable geographical differences; screening rates were lower for women in rural areas than for those in urban areas [24,25]. The high incidence of cervical cancer in several countries, including Ghana, is associated with a poor screening service use rate, which calls for focused initiatives to increase screening uptake and lessen the disease's effects [25,26]. The following screening methods are available: visual inspection with VIA and Enhanced Visual Assessment (EVA) mobile colposcopy; cytology (Pap) testing, and HPV DNA testing, but women had to pay out of pocket [27]. Ghana launched the HPV vaccine in October 2025. The Ghana Health Services, in collaboration with Vaccine Alliance (Gavi), WHO, and UNICEF, launched the country's first nationwide HPV vaccination campaign. The five-day campaign was expected to reach 2.4 million girls aged 9–14 years, both in and out of school, providing life-saving protection against cervical cancer [28].

Studies have found that HIV testing and having multiple sexual partners influenced cervical cancer screening [29–31]. However, in the Ghanaian context, there are limited studies that examine the interplay between multiple sexual partners and HIV testing on cervical cancer screening uptake. Existing studies in Ghana [32–36] have typically examined either sexual risk behaviours or HIV testing in isolation, therefore it is imperative to examine the influence of multiple sexual partners and HIV testing on cervical cancer screening uptake among women of reproductive age in Ghana.

## Materials and methods

### Data sources

The Ghana Statistical Service (GSS) conducted the 2022 Ghana Demographic and Health Survey (GDHS), which provided the data utilized in this study. Between October 17, 2022, and January 14, 2023, data were gathered. For the Demographic and Health Survey Program (DHS), GSS got technical support from the International Coaching Federation (ICF) to ensure that the survey procedures followed ethical research standards. The survey was funded by World Bank, The Global Fund, Korean International Cooperation Agency (KOICA), WHO, Government of Ghana, etc. Using data from the 2021 Population and Housing Census, Ghana Statistical Service adjusted the sample frame for the 2022 GDHS. Stratified two-stage cluster sampling was the sampling strategy employed in the 2022 GDHS. It was intended to produce representative findings for each of the 16 regions, for urban and rural areas, and at the national level. Using a probability proportional to size (PPS) approach, 618 target clusters were chosen for each region's urban and rural regions in the first stage. Following that, a systematic random selection of the clusters selected in the first phase was used to choose the target number of clusters with equal probability for both urban and rural regions in each region. The second step was listing and updating the households in each cluster. A sample of households was selected from this list. With assistance from ICF, 30 randomly selected households from each cluster were interviewed. A nationwide stratified representative sample of 18,450 households was included in the 2022 GDHS, from which 15,014 women of reproductive age (15–49 years) were interviewed.

### Measurements

**Outcome variable.** The outcome variable was cervical cancer screening. Eligible women, that is, women between 15–49 years old were asked, "Have you ever been examined for cervical cancer by a health worker?" The responses were categorized as "0" for "No" and "1" for "Yes".

**Explanatory variables.** HIV testing and multiple sexual partnership were the explanatory variables. The questions posed to women for HIV testing were "Have you ever been tested for HIV? The responses were coded "0" for "No" and "1" for "Yes". Multiple sexual partnerships were derived by asking women the number of sex partners excluding spouse in the last 12 months. The response was recoded as "0" = "No" and having one or more partners other than the spouse as "1" = "Yes".

**Covariates.** The study included ten covariates based on an extensive literature review and their availability in the data set [25,26,29,37–44]. The following covariates were included in the analysis: age was recoded (15–24, 25–34, 35–49), educational level (no education, primary education, secondary education, higher education), wealth index (poorest, poorer, middle, richer, richest), distance from health facility (big problem, not a big problem), frequency of reading newspaper or magazine (not at all, less than once a week, at least once a week), frequency of listening to radio (not at all, less than once a week, at least once a week), frequency of watching television (not at all, less than once a week, at least once a week), parity (0, 1–3, 4 and above), smokes cigarettes (no, yes), STI status (no, yes).

## Statistical analysis

The study utilized Stata version 14.0 to analyze data employing both descriptive and inferential statistics. Person's Chi-square test was used to determine whether there were any significant differences in cervical cancer screening uptake between the explanatory variables and covariates. Multivariable logistic regression analysis was used to examine the factors associated with cervical cancer uptake among women of reproductive age in Ghana. The multivariable logistics regression analyses consisted of two models. In model I, we adjusted for multiple sexual partnerships and HIV testing, and in the final model, we adjusted for other covariates. The data used for the analyses were weighted (v005/1000000) to account for the complex sampling of the survey. The results were reported as adjusted odds ratios (aOR) with their confidence intervals (CI) and statistical significance at p-value <0.05. A multicollinearity diagnostic test was performed to assess the variance inflation factors (VIF) for the variables that could account for cervical cancer screening (Min = 1.00, Mean VIF = 1.51, Max = 2.70). The rule of thumb was applied, where none of the variables had a higher VIF than required for exclusion in the multivariate analysis. All statistically significant variables were included in the final multivariate logistic regression analysis model.

## Ethical consideration

This study did not require ethical approval because we used secondary data from DHS, which sought ethical approval from the Ghana Nutrition and Health Research Institute (GNHRI) Review Board and the National Research Ethics Review Committee (NRERC) at the Ministry of Health before the data were collected. Informed consent was sought and obtained from all participants. Parental or guardian consent was sought for individuals below 18 years during the survey. The dataset is publicly available at the Measure DHS repository (https://dhsprogram.com/data/dataset/Ghana_Standard-DHS_2022.cfm?flag=1), and the report is available at https://dhsprogram.com/pubs/pdf/FR387/FR387.pdf.

## Results

### Participants sociodemographic characteristics

A total of 15,014 women of reproductive age were included in the study. From Table 1, the majority (35.8%) of women were in the 15–24 year age group, had secondary education (59.9%), and resided in the urban areas (57.0%). Approximately twenty-three percentage belonged to richer households, 77.7% reported that distance to a health facility was not a major problem, and 88.5% had never read a newspaper or magazine. However, 42.3% had listened to radio at least once a week, and 61.6% had watched television at least once a week. Most (41.2%) of women had 1–3 children, 77.0% did not have multiple sexual partners, 99.1% did not smoke cigarettes, 94.5% had never had any STI in the last 12 months, and 57.3% had ever tested for HIV.

**Table 1. Demographic characteristics of respondents on cervical cancer screening.**

| Variables | Categories | Weighted N (%) 15,014 | Cervical cancer screening | | p-value |
|---|---|---|---|---|---|
| | | | **No** | **Yes** | |
| **Mean Age** | 29.5 | | | | |
| **Age** | 15-24 | 5,376 (35.8) | 5,405 (98.2) | 99 (1.8) | <0.001 |
| | 25-34 | 4,592 (30.6) | 4,331 (93.9) | 283 (6.1) | |
| | 35-49 | 5,046 (33.6) | 4,588 (93.7) | 308 (6.3) | |
| **Education** | No education | 2,411 (16.1) | 3,263 (97.2) | 94 (2.8) | <0.001 |
| | Primary | 2,071 (13.8) | 2,176 (96.9) | 69 (3.1) | |
| | Secondary | 8,999 (59.9) | 7,808 (96.3) | 303 (3.7) | |
| | Higher | 1,533 (10.2) | 1,077 (82.8) | 224 (17.2) | |
| **Residence** | Urban | 8,557 (57.0) | 6,898 (93.7) | 464 (6.3) | <0.001 |
| | Rural | 6,457 (43.0) | 7,426 (97.1) | 226 (2.9) | |
| **Wealth Index Combined** | Poorest | 2,447 (16.3) | 3,599 (98.2) | 67 (1.8) | <0.001 |
| | Poorer | 2,712 (18.0) | 3,273 (97.2) | 93 (2.8) | |
| | Middle | 3,121 (20.8) | 2,881 (95.8) | 127 (4.2) | |
| | Richer | 3,379 (22.5) | 2,527 (94.1) | 159 (5.9) | |
| | Richest | 3,355 (22.4) | 2,044 (89.3) | 244 (10.7) | |
| **Distance to Health Facility** | Big problem | 3,354(22.3) | 3,800 (96.7) | 129 (3.3) | <0.001 |
| | Not a big problem | 11,660 (77.7) | 10,524 (94.9) | 561 (5.1) | |
| **Frequency of reading newspaper or magazine** | not at all | 13,293 (88.5) | 13,032 (95.8) | 571 (4.2) | <0.001 |
| | less than once a week | 1,182 (7.9) | 892 (91.9) | 79 (8.1) | |
| | at least once a week | 539 (3.6) | 400 (90.9) | 40 (9.1) | |
| **Frequency of listening to radio** | not at all | 4,993 (33.2) | 5,543 (97.0) | 173 (3.0) | <0.001 |
| | less than once a week | 3,674 (24.5) | 3,400 (95.4) | 163 (4.6) | |
| | at least once a week | 6,347 (42.3) | 5,381 (93.8) | 354 (6.2) | |
| **Frequency of watching television** | not at all | 3,463 (23.0) | 4,360 (97.9) | 94 (2.1) | <0.001 |
| | less than once a week | 2,305 (15.4) | 2,277 (95.7) | 102 (4.3) | |
| | at least once a week | 9,246 (61.6) | 7,687 (94.0) | 494 (6.0) | |
| **Parity** | 0 | 4,854 (32.3) | 4,528 (97.2) | 130 (2.8) | <0.001 |
| | 1-3 | 6,189 (41.2) | 5,714 (93.9) | 369 (6.1) | |
| | 4 and above | 3,970 (26.5) | 4,082 (95.5) | 191 (4.5) | |
| **Smoking** | No | 11,556 (77.0) | 14,220 (94.4) | 680 (4.6) | 0.033 |
| | Yes | 3,458 (23.0) | 104 (91.2) | 10 (8.8) | |
| **Multiple Sexual Partners** | No | 14,882 (99.1) | 11,304 (95.1) | 579 (4.9) | 0.002 |
| | Yes | 132 (0.9) | 3,020 (96.5) | 111 (3.5) | |
| **STI** | No | 14,191 (94.5) | 13,530 (95.5) | 632 (4.5) | 0.002 |
| | Yes | 823 (5.5) | 794 (93.2) | 58 (6.8) | |
| **HIV testing** | No | 6,403 (42.7) | 6,827 (98.4) | 109 (1.6) | <0.001 |
| | Yes | 8,611 (57.3) | 7,497 (92.8) | 581 (7.2) | |

## Bivariate analysis of factors associated with cervical cancer screening

From the Chi-square analysis, women's age, educational status, type of residence, household wealth status, distance to health facility, frequency of reading newspaper or magazine, frequency of listening to radio, frequency of watching television, parity, HIV testing and multiple sexual partners were significantly associated with cervical cancer screening (see Table 1).

**Prevalence of cervical cancer screening**

Fig 1 shows that only 5.0% [4.4–5.6] of women of reproductive age had ever screened for cervical cancer.

**Relational association between multiple sexual partners and HIV testing on cervical cancer screening**

Table 2 presents the association between multiple sexual partners and HIV testing on cervical cancer screening uptake. The results show that women with multiple sexual partners [cOR=0.69, 95% CI: 0.52–0.90] were less likely to screen for cervical cancer compared to those who do not have multiple sexual partners in the unadjusted model (model I). However, there was no association between multiple sexual partners and cervical cancer screening in the adjusted model (model II). Also, in model I, women who had ever tested for HIV [cOR=4.73, 95% CI:3.39–6.59] were more likely to screen for cervical cancer than those who had never tested. This was still significant after adjusting for covariates [aOR=2.47, 95% CI:1.61–3.80] in model II. With regard to the covariates, the probability of cervical cancer screening was higher among women aged 25−34 [aOR=2.10, 95% CI:1.47–3.00] and 35−49 years [aOR=3.17, 95% CI:2.16–4.66] than those aged 15−24 years. Also, women with higher level of education [aOR=2.99, 95% CI:1.75–5.10] were more likely to screen for cervical cancer than those with no form of education. Moreover, women who belonged to the middle [aOR=1.87, 95% CI:1.05–3.30] and the richest household wealth quintiles [aOR=1.90, 95% CI:1.08–3.35] were more likely to have cervical

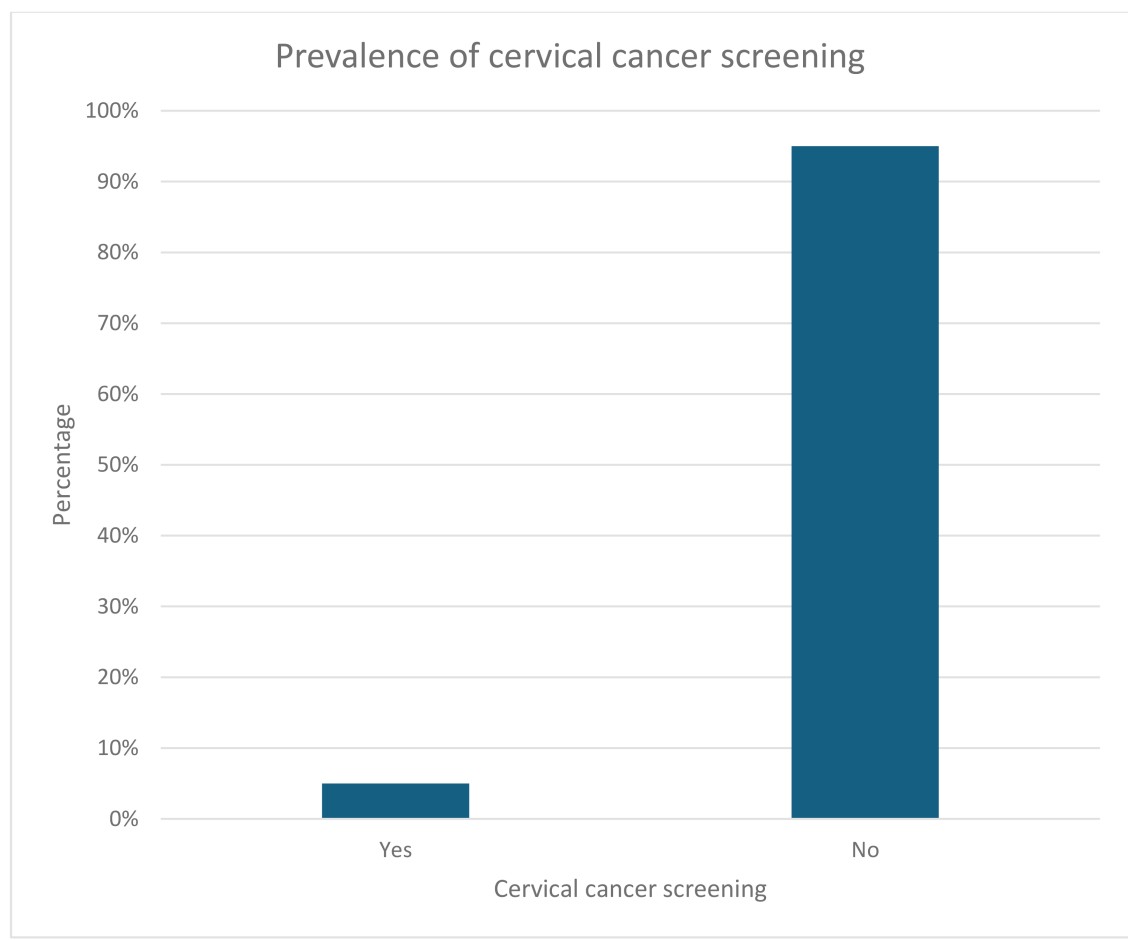

**Fig 1. Prevalence of Cervical Cancer Screening among Women.**

**Table 2. The relational association of multiple sexual partners and HIV testing on cervical cancer screening.**

| Variables | Model I cOR(95%CI) | Model II aOR(95%CI) |
|---|---|---|
| **Multiple Sexual Partners** | | |
| No | Ref. | Ref. |
| Yes | 0.69*[0.52-0.90] | 0.79[0.58-1.07] |
| **HIV testing** | | |
| No | Ref. | Ref. |
| Yes | 4.73***[3.39-6.59] | **2.47***[1.61-3.80]** |
| **Covariates** | | |
| **Age** | | |
| 15-24 | | Ref. |
| 25-34 | | **2.10***[1.47-3.00]** |
| 35-49 | | **3.17***[2.16-4.66]** |
| **Education** | | |
| No education | | Ref. |
| Primary | | 0.87[0.51- 1.49] |
| Secondary | | 0.97[0.63- 1.49] |
| Higher | | **2.99***[1.75-5.10]** |
| **Residence** | | |
| Urban | | Ref. |
| Rural | | 0.79[0.58-1.06] |
| **Wealth Index Combined** | | |
| Poorest | | Ref. |
| Poorer | | 1.30[0.84-2.01] |
| Middle | | **1.87*[1.05- 3.30]** |
| Richer | | 1.71[0.96-3.06] |
| Richest | | **1.90*[1.08-3.35]** |
| **Distance to Health Facility** | | |
| Big problem | | Ref. |
| Not a big problem | | 1.01[0.75-1.37] |
| **Frequency of reading newspaper or magazine** | | |
| not at all | | Ref. |
| less than once a week | | **1.47*[1.07-2.01]** |
| at least once a week | | 1.31[0.84-2.06] |
| **Frequency of listening to radio** | | |
| not at all | | Ref. |
| less than once a week | | 1.07[0.79-1.44] |
| at least once a week | | **1.31*[1.02-1.68]** |
| **Frequency of watching television** | | |
| not at all | | Ref. |
| less than once a week | | 1.28[0.86-1.89] |
| at least once a week | | 1.26[0.85-1.86] |
| **Parity** | | |
| 0 | | Ref. |
| 1-3 | | 1.16[0.82-1.64] |
| 4 and above | | 0.96[0.60-1.54] |

*(Continued)*

**Table 2.** (Continued)

| Variables | Model I<br>cOR(95%CI) | Model II<br>aOR(95%CI) |
|---|---|---|
| **STI** | | |
| No | | Ref. |
| Yes | | 1.26[0.83-1.92] |

**Legend:** aOR=Adjusted Odds ratio, CI = Confidence Interval; *p < 0.05, **p < 0.01, ***p < 0.001; Ref= Reference category.

cancer screening. Furthermore, women who read newspaper or magazine less than once a week [aOR=1.47, 95% CI:1.07–2.01] were more likely to have cervical cancer screening than those who did not read at all. Lastly, women who listened to radio at least once a week [aOR=1.31, 95% CI:1.02–1.68] also had higher odds of cervical cancer screening than those who did not listen to radio.

## Discussion

This study sought to examine the association between having multiple sexual partners, HIV testing, and cervical cancer screening uptake among women of reproductive age in Ghana. We found no association between having multiple sexual partners and cervical cancer screening; however, HIV testing was associated with higher odds of cervical cancer screening among women. We also observed higher odds of cervical cancer screening with increasing age, education, household wealth, and media exposure.

Contrary to previous evidence indicating a significant positive relationship between having multiple sexual partners and cervical cancer screening [29,44,45], our study found no such association. One possible reason could be that women who have sex with more than one partner may not think they are more likely to have cervical cancer, which might make them less motivated to seek cervical cancer screening [34]. Also, not knowing much about the relationship between high-risk sexual conduct and cervical cancer, together with the social stigma around both sexual behavior and reproductive health services, may make people less likely to seek health care. Additionally, the findings may be a result of structural barriers, such as limited access to screening services, lack of education on cervical cancer, and the cost of the service [34]. Our findings underscore the need for comprehensive public health interventions that integrate cervical cancer education into sexual and reproductive health programs, especially those targeting high-risk populations. The lack of a correlation between multiple sexual partners and cervical cancer screening in this study may indicate a confluence of cultural, behavioral, and structural influences. Cultural norms that inhibit candid discourse on sexual conduct may result in the underreporting of sexual partners, obscuring any genuine relationship. The limited understanding of the connection between sexual risk behaviors and cervical cancer suggests that sexual history may not significantly impact screening choices. Instead, women's screening habits are frequently influenced by factors such as fear, misunderstandings, low perceived vulnerability, and general health-seeking tendencies. Also, structural barriers such as access to screening services and cost further overshadow the individual behavioral influences [34].

Furthermore, the study found that women who had HIV testing were more likely to screen for cervical cancer. Our findings support the hypothesis that there is a significant association between HIV testing and cervical cancer screening uptake. One possible explanation is that HIV testing typically happens in healthcare institutions that simultaneously offer or encourage other preventative treatments, such as cervical cancer screening [46]. Women who have HIV tests may be more health-conscious or already involved with the healthcare system, which makes them more likely to hear about or be referred for cervical cancer screening. Our findings highlight the need for the integration of cervical cancer screening services into the routine HIV care as a gateway to broaden preventive health behaviors [47].

We found that increasing age was a predictor of cervical cancer screening uptake. Women aged between 35–49 years were more likely to screen for cervical cancer. This is in line with earlier studies that indicated that as women grow older, they tend to screen for cervical cancer [36,48–50]. This could be attributed to the fact that cervical cancer awareness is targeted towards older women and hence increasing its uptake compared to the younger ones, who do not benefit from these campaign activities [3]. Also, with increasing age, women become aware of the increased risk of certain diseases, including cervical cancer, which prompts them to screen for cervical cancer [3]. Consistent with previous studies [25,34,36,50–52], this study found that women with a higher level of education were more likely to screen for cervical cancer. A plausible explanation could be that educated women may be more exposed to and understand health information, including the relevance of screening for cervical cancer [53–56].

Also, our study found that women who belonged to the richest household wealth quintile were more likely to screen for cervical cancer. This is consistent with previous studies [25,29,57] which found a significant association between wealth and cervical cancer screening uptake. A possible explanation could be that wealthy women may be able to afford screening-related costs, such as transportation fares, the cost of screening, and related costs [57,58]. Lastly, mass media was found to be significantly associated with the uptake of cervical cancer, where women who listened to the radio at least once a week and read newspapers or magazines were more likely to screen for cervical cancer. This resonates with earlier studies that indicated a positive correlation between mass media exposure and cervical cancer screening [25]. This could be that relevant health information is published in newspapers or magazines or broadcast via the radio, which increases the knowledge and awareness for the prevention of cervical cancer screening [59].

## Policy implications

To increase cervical cancer screening uptake, several targeted interventions should be considered among women of reproductive age. The integration of cervical cancer screening with HIV services can thereby simplify access, reduce missed opportunities, and enable women already in contact with the HIV care system to be screened during routine visits. Health education programs in schools for adolescents and young women could help to promote early concept familiarity on cervical carcinoma, HPV diagnosis, and regular screening. Furthermore, mobile screening clinics, allow services to reach underprivileged and hard-to-access communities, which in turn eliminates geographic and financial barriers. These interventions, in conjunction with community health worker outreach, culturally appropriate education, and improved referral systems, will lead to a substantial increase in screening uptake and overall cervical cancer prevention for women.

## Strengths and limitations

The findings may be generalized to the larger population of Ghanaian women of reproductive age, given the use of a nationally representative dataset. The study's strength stems from its originality as the first to investigate the relationship between multiple sexual partners and HIV testing on cervical screening uptake in Ghana. However, the study has limitations. First, variables used in the study were all self-reported, potentially introducing recall and social desirability biases. However, we do not expect systematic differences in recall and social biases for sexual behavior and HIV testing among the study participants. Second, the use of a cross-sectional design cannot infer causality. Lastly, though WHO recommends screening for cervical cancer at age 25, especially with individuals with HIV, the study included participants aged 15–24 years old. The findings may have been affected by potential misclassification arising from stigma-related underreporting of sexual behaviors, as well as the inclusion of participants aged 15–24 years who fall below the recommended age for cervical cancer screening."

## Conclusion

Despite the strong evidence that cervical cancer can be prevented by early screening to detect precancerous lesions, the utilization of cervical cancer screening tests remains low in Ghana. Though the study found no significant association

between multiple sexual partnership and cervical screening uptake, HIV testing was a predictor of cervical cancer screening among women in Ghana. Therefore, public health initiatives are encouraged to educate all women, regardless of their sexual behavior and HIV status, on the significance of cervical cancer screening. This may promote the uptake of cervical cancer screening among women in Ghana.

## Supporting information

**S1 File. A forest plot for multivariate result.**
(DOCX)

## Author contributions

**Conceptualization:** Agani Afaya, Wise Awunyo, Mary Adaeze Udeoha, Matilda Mawusi Kodjo, Maxwell Tii Kumbeni.

**Data curation:** Agani Afaya, Wise Awunyo.

**Formal analysis:** Agani Afaya, Wise Awunyo.

**Investigation:** Agani Afaya, Wise Awunyo, Mary Adaeze Udeoha, Matilda Mawusi Kodjo, Maxwell Tii Kumbeni.

**Methodology:** Agani Afaya, Wise Awunyo, Mary Adaeze Udeoha, Matilda Mawusi Kodjo, Maxwell Tii Kumbeni.

**Project administration:** Agani Afaya.

**Software:** Agani Afaya, Wise Awunyo.

**Supervision:** Agani Afaya, Maxwell Tii Kumbeni.

**Validation:** Agani Afaya, Wise Awunyo, Maxwell Tii Kumbeni.

**Visualization:** Agani Afaya.

**Writing – original draft:** Agani Afaya, Wise Awunyo, Mary Adaeze Udeoha, Matilda Mawusi Kodjo, Maxwell Tii Kumbeni.

**Writing – review & editing:** Agani Afaya, Wise Awunyo, Mary Adaeze Udeoha, Matilda Mawusi Kodjo, Maxwell Tii Kumbeni.

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
