## [Decision Letter · Decision Letter 0]

13 Oct 2025

Thank you for submitting your manuscript to PLOS ONE. After careful consideration, we feel that it has merit but does not fully meet PLOS ONE’s publication criteria as it currently stands. Therefore, we invite you to submit a revised version of the manuscript that addresses the points raised during the review process.

We look forward to receiving your revised manuscript.

Kind regards,

Minal Dakhave

Academic Editor

PLOS ONE

Journal Requirements:

3. We note you have included a table to which you do not refer in the text of your manuscript. Please ensure that you refer to Table 2 in your text; if accepted, production will need this reference to link the reader to the Table.

**Additional Editor Comments:**

**General Assessment:**

The manuscript addresses an important public health topic—the influence of multiple sexual partners and HIV testing on cervical cancer screening among women of reproductive age in Ghana. The study is relevant and uses a nationally representative dataset. However, major revisions are needed to improve clarity, organization, scientific rigor, and readability.

**Specific Comments:**

**Introduction:**

**Redundancy & Flow:**  Merge repeated global cervical cancer statistics; present global → regional (SSA) → Ghana sequentially.**WHO Strategy:**  Provide context on progress toward the 90-70-90 targets, especially in LMICs/SSA, with recent references (2023–2024).**Context for Ghana:**  Integrate fragmented national data into a cohesive paragraph; include screening programs, HPV vaccination, and HPV DNA testing availability. Use recent national surveys (e.g., GLOBOCAN 2022).**Study Rationale:**  Clarify gaps in Ghana-specific data and how this study addresses them. Example: emphasize limited prior work on HIV testing and sexual behavior interactions in Ghana.**Language & Style:**  Simplify long/redundant sentences, maintain consistent tense, and adopt academic terminology (e.g., “age-standardized incidence rate”). Avoid starting sentences with numerical data.**Referencing:**  Verify all citations, update with recent literature, and ensure numerical references are accurate.**Structure:**  Suggested flow: Global burden → SSA burden → Ghana → Screening & WHO strategy → Barriers → Study rationale

**Results:**

**Clarity:**  Summarize key trends instead of repeating table data.**Tables:**  Reorganize, align columns, label models clearly, and highlight significant results.**Integration:**  Separate descriptive and inferential results with subheadings.**Non-significant Variables:**  Interpret or explicitly note lack of association.**Visual Aids:**  Consider bar charts for prevalence and forest plots for multivariable results.

**Discussion:**

**Non-significant Findings:**  Expand discussion on cultural, behavioral, or structural factors influencing the lack of association between multiple sexual partners and screening.**Policy Implications:**  Suggest interventions, e.g., integrating cervical screening with HIV services, school programs, or mobile clinics.**Limitations:**  Specify potential misclassification due to stigma and inclusion of participants below recommended screening age (15–24 years) as factors possibly affecting results

**Conclusion:**

The study highlights low cervical cancer screening uptake in Ghana and identifies HIV testing as a significant predictor. Strengthening the discussion of context-specific factors and practical interventions will enhance the manuscript’s relevance and impact.

**Recommendation:**  Major revision required.

Reviewers' comments:

Reviewer's Responses to Questions

**Comments to the Author**

1. Is the manuscript technically sound, and do the data support the conclusions?

Reviewer #1: Partly

2. Has the statistical analysis been performed appropriately and rigorously?

Reviewer #1: No

3. Have the authors made all data underlying the findings in their manuscript fully available?

Reviewer #1: No

4. Is the manuscript presented in an intelligible fashion and written in standard English?

Reviewer #1: Yes

Reviewer #1: Thank you for the opportunity to review the study entitled “The relational association between multiple sexual partners and HIV testing on cervical cancer screening among women of reproductive age in Ghana.” Many thanks to the authors for doing this work. I have a few comments and suggestions that may help strengthen the paper.

Introduction

The paper examines the relationship between multiple sexual partners, HIV and cervical cancer screening. However, no background information on the burden of cervical cancer in Ghana is provided which will better situate the study.

Since the focus is on the association between multiple sexual partners, HIV and cervical cancer screening, it would be useful to provide supporting literature on these relationships in the background section.

Lines 111–115: The authors have not made a sufficiently strong case for why this study is necessary. This section should be expanded to highlight the research gap and rationale.

Methods

Lines 141–150: Both the outcome and exposure variables were self-reported. This introduces the potential for recall and reporting bias, which should be acknowledged as a limitation.

Lines 164–172: Several indicators were adjusted for in the multivariable logistic regression model. How were these potential confounders identified? Please clarify the criteria used for their inclusion.

Results

Lines 191–199: The results section should begin by stating the total number of women included in the study sample.

Lines 191–199: What was the mean or median age of the participants? This should be provided as part of the descriptive results.

Lines 191–199: Percentages were reported in the text, but absolute numbers should also be included for clarity.

Table 2: The heading and content should be revised. This is not an “association” table. Also, column percentages for each indicator should be included.

Lines 206–210: Model 1 is the unadjusted model, while Model 2 is the adjusted model. This distinction should be clearly stated in the text.

Table 3: Two additional columns should be added to show the distribution of the outcome variable across the exposures.

Table 3: Results for some variables are missing in Model 1. These should be included to allow readers to see how the odds ratios change after adjustment.

Discussion

The discussion is too general. Given that the study focuses on the relationship between multiple sexual partners, HIV testing, and cervical cancer screening in Ghana, the discussion should highlight contextual factors specific to Ghana that may explain the findings.

Lines 285–292: Since this is a cross-sectional study, causal inference cannot be made. However, the authors should indicate specific features of the study (eg., study population or unmeasured confounders) that may have influenced the results.

The implications of the findings for cervical cancer screening programs in Ghana are not clear. The authors should expand this section to discuss how the study contributes to policy, practice or future interventions.

**Do you want your identity to be public for this peer review?** For information about this choice, including consent withdrawal, please see our Privacy Policy

Reviewer #1: No

---

## [Author Response · Author response to Decision Letter 1]

3 Dec 2025

Dear Editor,

We would like to thank you sincerely for the insightful reviewer comments on our manuscript and for the opportunity to resubmit the manuscript for a second round of review and further publication consideration. We find the comments very useful and have responded to them to the best of our knowledge. We acknowledge that the comments have no doubt helped improve the quality of our manuscript.

We herein provide responses to detail how we have addressed the comments of the reviewers’ point-by-point. For easy identification, the reviewers’ comments have been repeated while Authors’ responses appear in Bold text.

General Assessment:

The manuscript addresses an important public health topic—the influence of multiple sexual partners and HIV testing on cervical cancer screening among women of reproductive age in Ghana. The study is relevant and uses a nationally representative dataset. However, major revisions are needed to improve clarity, organization, scientific rigor, and readability.

Specific Comments:

Introduction:

1. Redundancy & Flow: Merge repeated global cervical cancer statistics; present global → regional (SSA) → Ghana sequentially.

Response: We have addressed the above comment within the background

2. WHO Strategy: Provide context on progress toward the 90-70-90 targets, especially in LMICs/SSA, with recent references (2023–2024).

Response: We included this in the background as recommended

3. Context for Ghana: Integrate fragmented national data into a cohesive paragraph; include screening programs, HPV vaccination, and HPV DNA testing availability. Use recent national surveys (e.g., GLOBOCAN 2022).

Response: we have provided the screening methods in Ghana and the availability HPV DNA testing in the background

4. Study Rationale: Clarify gaps in Ghana-specific data and how this study addresses them. Example: emphasize limited prior work on HIV testing and sexual behavior interactions in Ghana.

Response: This has been indicated in the last paragraph

5. Language & Style: Simplify long/redundant sentences, maintain consistent tense, and adopt academic terminology (e.g., “age-standardized incidence rate”). Avoid starting sentences with numerical data.

Response: The above comments have been addressed as suggested.

6. Referencing: Verify all citations, update with recent literature, and ensure numerical references are accurate.

Response: Thank you, we have updated most of the references used in the study.

7. Structure: Suggested flow: Global burden → SSA burden → Ghana → Screening & WHO strategy → Barriers → Study rationale

Response: The background was restructured as suggested above.

Results:

1. Clarity: Summarize key trends instead of repeating table data.

RESPONSE: We have summarized the key findings in the results section.

2. Tables: Reorganize, align columns, label models clearly, and highlight significant results.

RESPONSE: We have revised the models clearly and highlighted significant results

3. Integration: Separate descriptive and inferential results with subheadings.

RESPONSE: This has been done.

4. Non-significant Variables: Interpret or explicitly note lack of association.

RESPONSE: This has been explained within the results section

5. Visual Aids: Consider bar charts for prevalence and forest plots for multivariable results.

RESPONSE: Thank you, we have considered bar charts for prevalence and the forest plots for multivariable results. But it seems the table is more visible

Discussion:

1. Non-significant Findings: Expand discussion on cultural, behavioral, or structural factors influencing the lack of association between multiple sexual partners and screening.

RESPONSE: We have expanded on the discussion, focusing on cultural, behavioral, or structural factors influencing the lack of association between multiple sexual partners and screening

2. Policy Implications: Suggest interventions, e.g., integrating cervical screening with HIV services, school programs, or mobile clinics.

RESPONSE: We have addressed this

3. Limitations: Specify potential misclassification due to stigma and inclusion of participants below recommended screening age (15–24 years) as factors possibly affecting results

Conclusion:

The study highlights low cervical cancer screening uptake in Ghana and identifies HIV testing as a significant predictor. Strengthening the discussion of context-specific factors and practical interventions will enhance the manuscript’s relevance and impact.

REVIEWER ONE

Reviewer #1: Thank you for the opportunity to review the study entitled “The relational association between multiple sexual partners and HIV testing on cervical cancer screening among women of reproductive age in Ghana.” Many thanks to the authors for doing this work. I have a few comments and suggestions that may help strengthen the paper.

Introduction

The paper examines the relationship between multiple sexual partners, HIV and cervical cancer screening. However, no background information on the burden of cervical cancer in Ghana is provided, which will better situate the study.

RESPONSE: Thank you for this recommendation. We have therefore revised the background to include cervical cancer burden.

Since the focus is on the association between multiple sexual partners, HIV, and cervical cancer screening, it would be useful to provide supporting literature on these relationships in the background section.

RESPONSE: actually there are limited studies in Ghana that have assessed the relationship, but we have provided justification for this study.

Lines 111–115: The authors have not made a sufficiently strong case for why this study is necessary. This section should be expanded to highlight the research gap and rationale.

RESPONSE:

We have included the research gap

Methods

Lines 141–150: Both the outcome and exposure variables were self-reported. This introduces the potential for recall and reporting bias, which should be acknowledged as a limitation.

RESPONSE: This is already under the limitation section but we have still revised it.

Lines 164–172: Several indicators were adjusted for in the multivariable logistic regression model. How were these potential confounders identified? Please clarify the criteria used for their inclusion.

RESPONSE: We reviewed literature to identify the potential confounders and have referenced these studies

Results

Lines 191–199: The results section should begin by stating the total number of women included in the study sample.

RESPONSE: This has been included in the first sentence under the results section.

Lines 191–199: What was the mean or median age of the participants? This should be provided as part of the descriptive results.

RESPONSE: We used the categorical variable, and it will be a form of repeating, which is why we could not include it.

Lines 191–199: Percentages were reported in the text, but absolute numbers should also be included for clarity.

RESPONSE We have included the percentages

Table 2: The heading and content should be revised. This is not an “association” table. Also, column percentages for each indicator should be included.

Response: This has been revised

Lines 206–210: Model 1 is the unadjusted model, while Model 2 is the adjusted model. This distinction should be clearly stated in the text.

RESPONSE: This has been corrected

Table 3: Two additional columns should be added to show the distribution of the outcome variable across the exposures.

RESPONSE: This has been shown in Table 1

Table 3: Results for some variables are missing in Model 1. These should be included to allow readers to see how the odds ratios change after adjustment.

RESPONSE: We used chi-square analysis to determine the level of association for all the variables before adjusting for them in model II. This format is also acceptable as unadjusted odds and chi-square analysis are all bivariate in nature

Discussion

The discussion is too general. Given that the study focuses on the relationship between multiple sexual partners, HIV testing, and cervical cancer screening in Ghana, the discussion should highlight contextual factors specific to Ghana that may explain the findings.

RESPONSE: We have addressed these concerns

Lines 285–292: Since this is a cross-sectional study, causal inference cannot be made. However, the authors should indicate specific features of the study (eg., study population or unmeasured confounders) that may have influenced the results.

RESPONSE:

The implications of the findings for cervical cancer screening programs in Ghana are not clear. The authors should expand this section to discuss how the study contributes to policy, practice or future interventions.

RESPONSE

This section has been included.

---

## [Decision Letter · Decision Letter 1]

16 Feb 2026

The relational association between multiple sexual partners and HIV testing on cervical cancer screening among women of reproductive age in Ghana: a national population analysis

PONE-D-25-39510R1

Dear Dr. Wise Awunyo,

We’re pleased to inform you that your manuscript has been judged scientifically suitable for publication and will be formally accepted for publication once it meets all outstanding technical requirements.

Kind regards,

Minal Dakhave

Academic Editor

PLOS One

Additional Editor Comments (optional):

Reviewers' comments:

Reviewer's Responses to Questions

**Comments to the Author**

Reviewer #1: All comments have been addressed

2. Is the manuscript technically sound, and do the data support the conclusions?

Reviewer #1: Yes

3. Has the statistical analysis been performed appropriately and rigorously?

Reviewer #1: Yes

4. Have the authors made all data underlying the findings in their manuscript fully available?

Reviewer #1: No

5. Is the manuscript presented in an intelligible fashion and written in standard English?

Reviewer #1: Yes

Reviewer #1: Authors have addressed all my previous comments on the paper. I do not have any additional comment.

**Do you want your identity to be public for this peer review?** For information about this choice, including consent withdrawal, please see our Privacy Policy

Reviewer #1: No

---

## [Editor Report · Acceptance letter]

PONE-D-25-39510R1

PLOS One

Dear Dr. Awunyo,

I'm pleased to inform you that your manuscript has been deemed suitable for publication in PLOS One. Congratulations! Your manuscript is now being handed over to our production team.

Kind regards,

on behalf of

Dr. Minal Dakhave

Academic Editor

PLOS One